# Anatomical Longitudinal Cortical Surface Registration

**Aakash Saboo**[1]                                       AAKASH.SABOO@KCL.AC.UK
[1] *Kings College London*
**Ashleigh Davies**[1]                                 ASHLEIGH.DAVIES@KCL.AC.UK
**Nashira Baena**[1]                                   NASHIRA.BAENA@KCL.AC.UK
**Kaili Liang**[1]                                       KAILI.LIANG@KCL.AC.UK
**Jiaxin Xiao**[1]                                       JIAXIN.1.XIAO@KCL.AC.UK
**Yourong Guo**[1]                                     YOURONG.GUO@KCL.AC.UK
**Renato Basenczi**[1]                                 RENATO.BESENCZI@KCL.AC.UK
**Jonathan O'Muircheartaigh**[1]                        JONATHANOM@KCL.AC.UK
**Emma Robinson** [1]                                 EMMA.ROBINSON@KCL.AC.UK

**Editors:** Accepted for publication at MIDL 2026

## Abstract

Longitudinal cortical surface registration is essential for accurately characterizing developmental and neurodegenerative trajectories, thereby facilitating a mechanistic understanding of cortical growth and the identification of biomarkers. This is hindered by current registration networks, which works on spherical projections of the cortical surface. Therefore, In this work, we present a novel longitudinal registration framework that operates directly on complex anatomical geometries by integrating a learning-based network with pairwise instance optimization. This hybrid strategy leverages the network to establish a robust initial alignment, which is subsequently refined through optimization to ensure high-fidelity registration. We demonstrate that this method yields growth maps with superior smoothness compared to baselines, enhancing their clinical utility, while rigorously preserving topological integrity as evidenced by analyses of self-intersecting faces, areal distortion, and anisotropic strain; source code is available on this github repository.

**Keywords:** Geometric Deep Learning, Surface Registration,

## 1. Introduction

Two key sources of variability confound the search for neuroimaging biomarkers: natural heterogeneity of brain shape (Guo et al., 2023, 2025; Van Essen, 2004; Mangin et al., 2004) and inter-individual differences in disease progression (Marquand et al., 2016). Therefore, longitudinal studies have gained interest due to their ability to increase statistical power through comparing rates of change rather than absolute values of features (Sauty and Durrleman, 2023; da Silva et al., 2025; Garcia et al., 2018b; Urru et al., 2025; Chauveau et al., 2021; Fiford et al., 2017; Hazlett et al., 2020; Huang et al., 2022; Wang et al., 2019). Accurate measurement of temporal differences, however, requires anatomically precise and biomechanically informed longitudinal surface registration to track morphological changes effectively. This is particularly challenging during the fetal and neonatal periods, which are characterized by rapid growth and cortical folding.

Inter-subject cortical alignment (learning point-to-point correspondence) is a difficult problem since human brains fold in very different ways – generating very different cortical shapes for which true spatial correspondences are unknown. Inter-subject alignment

is therefore an approximation problem where this mapping problem needs to be highly-regularised to discourage "biologically-implausible" solutions. The most successful frameworks simplify the problem by projecting cortical anatomies to spheres over which summary metrics, such as sulcal depth or curvature, may be matched (Fischl et al., 1999; Yeo et al., 2009; Robinson et al., 2014, 2017; Besenczi et al., 2024; Ren et al., 2024; Suliman et al., 2023; Zhao et al., 2021; Zhang et al., 2024; Asuliman, 2022; Zhao et al., 2024; Cheng et al., 2020). However, spherical projection introduces distortions (rescaling of vertex neighbourhoods), limiting the biological interpretability of these warps. While efforts have been made towards inferring deformations of cortical anatomy from these mappings. (Robinson et al., 2017; Yuan et al., 2024), it loses connection with the 3D ambient space (see Section 2). Without learning 3D mappings there is no principled way to learn to temporally interpolate between the two timepoints. In our case, exact correspondence between surfaces (acquired from the same individual at different gestational time points) is much easier to define but the deformations required are much more extreme, time varying and governed by physical laws of soft tissue mechanics. For this there is a need of lightweight, flexible framework that can encode shape but output deformation fields in 3D, providing an avenue to prescribe biomechanics of the deformations for forward modelling. The objective of this work is conversely to lay down the foundations of a future forward model of cortical growth and folding.

Within the shape modeling community, shape analysis has been done through geometric deep learning solutions such as Graph based methods (Li et al., 2025, 2022; Besson et al., 2020; Kwon et al., 2025; Ripart et al., 2025), Transformers (Cheng et al., 2022; Dahan et al., 2022), MeshCNN (Hanocka et al., 2019) and SubdivNet (Hu et al., 2022), and DiffusionNet (Sharp et al., 2020) which offers specific advantages in terms of its discretisation invariance and flexible learning framework. Recently, optimization based deep-learning models have gained significant interest for image registration applications (Jensen et al., 2022; Lowes et al., 2025; Banús et al., 2025; Zimmer et al., 2024; Tian et al., 2023). These models leverage instance optimization methods to achieve better performance by iteratively refining the deformation field, often resulting in more accurate and robust registrations compared to purely feed-forward approaches

**Contributions** This paper brings together Shape Modeling and Image Registration and presents a novel learning-based Longitudinal Cortical Surface Registration framework for smooth 3D warping of complex cortical anatomies. Focusing specifically on the challenging problem of perinatal cortical growth and folding, which is characterised by dramatic growth and large deformations (da Silva et al., 2025; Garcia et al., 2018b), we learn a biologically plausible deformation field that aligns the cortical surfaces at any two time points. This is achieved by leveraging discretisation invariant DiffusionNet (Sharp et al., 2020) modules and combining population-based training with instance optimisation. Furthermore, this work lays a foundation by establishing a principled framework to learn deformations in the 3D ambient space, for prescribing the biomechanics of deformations for forward modeling/interpolate between two time points.

## 2. Related works

The problem of learning-based modelling of cortical growth or atrophy has so far largely been tackled through template deformation approaches (Bongratz et al., 2025; Wu et al.).

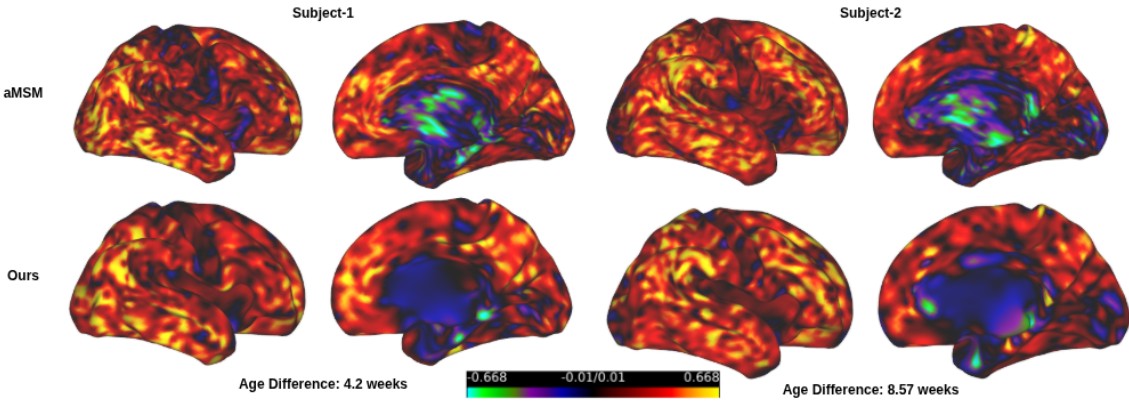

Figure 1: Growth maps

These benefit from the learning power of 3D convolutions but prioritize surface reconstruction over vertex correspondence and ultimately lack any clear framework for prescribing the biomechanics of the process, which is constrained relative to the surface geometry. Methods such as V2C-Long (Bongratz et al., 2025) show clear advantages for modelling neurodegeneration, where spatial changes are subtle and gradual but no framework has yet been able to capture the non-uniform dynamics of fetal and neonatal cortical folding, which occur at different rates across the surface.

Anatomical multimodal surface matching (aMSM) (Robinson et al., 2017) represents an example of a classical solution that incorporates biomechanically-inspired, hyperelastic, strain-energy penalties on growth tangential to the cortical surface. Experiments using aMSM to study perinatal cortical development have shown that it captures expected growth trends and delivers insight into differential patterns typical of preterm and fetal development (da Silva et al., 2025; Garcia et al., 2018a). However, at its source aMSM is a spherical alignment method that approximates the anatomical warp using the inherent vertex correspondence of white-matter (WM) and spherical meshes (Makropoulos et al., 2018; Fischl, 2012; Ma et al., 2025) to interpolate the deformed sphere onto the target anatomy. In this paper we therefore seek to develop a learning-framework that preserves the biological modelling power of aMSM but learns individually tailored deformations that smoothly warp the 3D anatomical (white matter) mesh.

## 3. Methods

Given two surface meshes $M_A = \{\mathbf{v}_i^A \in \mathbb{R}^3\}_{i=1}^{N_A}$ and $M_B = \{\mathbf{v}_i^B \in \mathbb{R}^3\}_{i=1}^{N_B}$, our goal is to find a mapping $\varphi$ such that $\varphi \circ M_A \approx M_B$ - i.e. shapes that approximately overlap while the mapping remains smooth and biologically plausible.

In this paper, $\varphi$ is parameterised as a neural network (NN) trained over a set of longitudinal mesh pairs $\mathcal{M} = \{(M_i^A, M_i^B)\}_{i=1}^N$. To balance generalisation and precision we optimise the framework in two stages: 1) a global learning-based framework that learns approximate mappings by training a DiffusionNet (Sharp et al., 2020) to learn vertex-wise

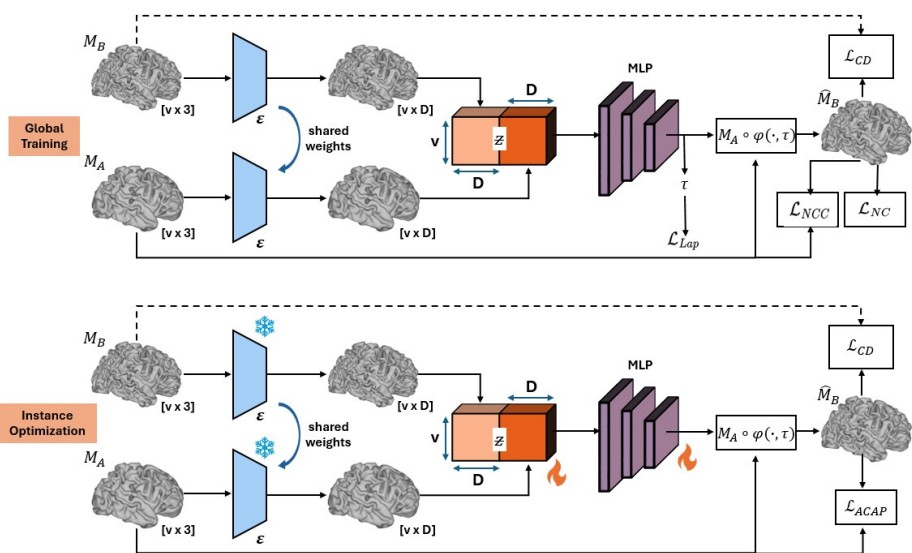

Figure 2: Model architecture. The first row shows the global training phase. The second row shows the instance optimization where the encoders are frozen, but the latent vectors $\mathcal{Z}$ and the MLP are optimized iteratively.

shape features; followed by 2) per-pair instance optimisation to refine high-frequency folding patterns. An overview of the architecture is shown in Fig. 2.

### 3.1. DiffusionNet feature extraction

In both networks the encoder $\mathcal{E}_{\theta_1}$ utilizes DiffusionNet (Sharp et al., 2020) to extract per-vertex features. DiffusionNet works by using the Laplace-Beltrami Operator ($\Delta$), which encodes the shape harmonics of each mesh, to learn a diffusion process over the surface. This allows the network to aggregate information from local and distant regions without relying on mesh triangulation. In practice, it uses the discrete cotangent Laplacian ($\mathbf{L}$), unique to each mesh, which weights its adjacency matrix relative to the actual geometry (edge lengths and angles) of the mesh; this is then normalised with a Mass Matrix ($\mathbf{M}$): a diagonal matrix that accounts for the area/volume associated with each vertex. These arrays are passed to a diffusion layer which applies the heat equation $\frac{\partial \mathbf{u}_t}{\partial t} = \Delta \mathbf{u}_t$, for a given scalar field $\mathbf{u}$. In practice, the solution to this equation is discretised as: $\mathbf{u}_t = \mathbf{e}^{-tL}\mathbf{u}_0$, with $\mathbf{u}_0$ representing the initial condition (or input feature matrix). This operation is computed efficiently using Spectral Acceleration (using the eigenvalues/eigenvectors of the mesh Laplacian). The objective is to learn a unique diffusion time $t$ for each feature channel, to return per-vertex features representations that encode shape at different scales. Since pure diffusion is isotropic, DiffusionNet also incorporates spatial gradient of the features (dot products of feature gradients), which gives the network the ability to detect directional patterns, similar to edge detectors in image CNNs. These features (learnt separately for each vertex) are then passed to MLPs for non-linear transformation of the layer. The key

advantages of DiffusionNet is therefore that the process can be generalised in a rotation-invariant, discretization-agnostic way through applying a ($\mathbf{L}$), unique to each mesh, but generalising the learnt (per-channel) diffusion time across samples. In practice, we use 4 DiffusionNet blocks in our encoder.

### 3.2. Global Alignment

Deformation network $\varphi_{\theta_G}$ is first trained over a population of longitudinal surface pairs to learn an approximate (coarse) deformation such that the deformed surface ($\varphi \circ M_A$) maintains alignment/vertex correspondence with the original shape while being as close in shape as possible to the target surface ($M_B$). This network consists of an MLP built from three hidden layers with 512 units each, utilizing ReLU activations for hidden layers and a linear output for the three-dimensional deformation field. This is optimised using a Chamfer distance ($\mathcal{L}_{\text{sim}}$) that drives the deformed source towards the target $M_B$, a Normalized Cross-Correlation ($\mathcal{L}_{\text{NCC}}$) regularization, employed to maintain vertex correspondence between the deformed state and the original $M_A$. $\mathcal{L}_{\text{Lap}}$ is the Laplacian regularization applied to the estimated displacement field ($\tau$) to encourage smooth deformations, and $\mathcal{L}_{\text{NC}}$ is the Normal Consistency regularization to ensure smoothness of the resultant surface. To select the hyperparameters, we performed an empirical evaluation on a subset of the data, evenly spanning gestational age, comprising 14 samples for validation and 13 for testing. We trained multiple models with: $\lambda_{\text{sim}} \in 10^3, 10^4, 10^5, \lambda_{\text{NCC}} \in 1, 10, 100, \lambda_{\text{lap}} \in 1, 10, 100, \lambda_{\text{NC}} \in 1, 10, 100$. Based on achieving the lowest reconstruction error, measured by Chamfer Distance (CD), and the highest alignment, measured by Normalized Cross-Correlation (NCC), we selected $\lambda_{\text{sim}} = 10^4$, and $\lambda_{\text{NCC}} = \lambda_{\text{lap}} = \lambda_{\text{NC}} = 10$.

$$\theta^* = \underset{\theta_1, \theta_2}{\operatorname{argmin}} \frac{1}{N} \sum_{i=1}^{N} \mathcal{L}(M_i^A, M_i^B, \tau_i) \tag{1}$$

$$\mathcal{L}(M_A, M_B, \tau) = \mathcal{L}_{\text{sim}}(M_A \circ \varphi, M_B) + \lambda_{\text{NCC}} \mathcal{L}_{\text{NCC}}(M_A \circ \varphi, M_A) \\ + \lambda_{\text{Lap}}(\tau) + \lambda_{\text{NC}}(M_A \circ \varphi) \tag{2}$$

### 3.3. Instance Optimisation

Due to the considerable heterogeneity of cortical shapes, limited longitudinal data and the ill-posed nature of shape matching, generalisable mapping is difficult to learn. For these reasons we employ an instance-specific optimization (fitting) scheme to refine registration for each pair. Let $\theta_1^*$ and $\theta_2^*$ denote the pre-trained parameters of the encoder and deformation network, respectively. For a specific pair of meshes ($M_A, M_B$), we first extract the combined latent representation $\mathcal{Z}$ using the fixed encoder $\mathcal{E}_{\theta_1^*}$. We then treat the latent code $\mathcal{Z}$ and the deformation network parameters $\theta_2$ as learnable variables to further optimize the deformation field $\tau$. The optimization objective is defined as:

$$\{\mathcal{Z}^*, \theta_2^{**}\} = \underset{\mathcal{Z}, \theta_2}{\operatorname{argmin}} \left(\mathcal{L}_{\text{sim}}(M_A \circ \varphi, M_B) + \lambda_{\text{ACAP}} \mathcal{L}_{\text{ACAP}}(M_A \circ \varphi, M_A)\right) \tag{3}$$

In this stage, $\mathcal{L}_{\text{sim}}$ remains the Chamfer distance. $\mathcal{L}_{\text{ACAP}}$ represents the As-Conformal-As-Possible energy. Given that the cortex grows in size (increasing surface area and volume),

the mesh faces must stretch within specified limits while preserving local geometry. The ACAP regularization facilitates this by allowing for isotropic scaling (simulating growth) while minimizing angular distortion, thereby preserving the intrinsic anatomical shape features. The ACAP energy minimizes the squared deviation between the deformed edge vectors $\mathbf{e}'_{ij}$ and the rest edge vectors $\mathbf{e}_{ij}$, transformed by the averaged similarity transformation of their endpoints. The total energy is defined as:

$$E_{\text{ACAP}} = \sum_{(i,j) \in E} \left\| \mathbf{e}'_{ij} - \frac{1}{2}(s_i \mathbf{R}_i + s_j \mathbf{R}_j)\mathbf{e}_{ij} \right\|^2 \tag{4}$$

where: $E$ is the set of all edges in the mesh, $\mathbf{e}_{ij} = \mathbf{p}_j - \mathbf{p}_i$ is the edge vector in the rest pose. $\mathbf{e}'_{ij} = \mathbf{p}'_j - \mathbf{p}'_i$ is the edge vector in the deformed pose. $\mathbf{R}_i \in SO(3)$ is the optimal rotation for vertex $i$, $s_i \in \mathbb{R}^+$ is the optimal uniform scaling factor for vertex $i$. For each vertex $i$, the optimal rotation $\mathbf{R}_i$ and scaling factor $s_i$ are derived from the local weighted covariance matrix of its 1-ring neighborhood $\mathcal{N}(i)$:

$$S_i = \sum_{j \in \mathcal{N}(i)} w_{ij} \mathbf{e}'_{ij} \mathbf{e}_{ij}^T \tag{5}$$

where $w_{ij}$ are the cotangent weights. By performing the Singular Value Decomposition (SVD) of the covariance matrix, $S_i = U_i \Sigma_i V_i^T$, we obtain the optimal parameters. The rotation is extracted via the polar decomposition factor of the covariance matrix:

$$\mathbf{R}_i = \mathbf{U}_i \mathbf{V}_i^T \tag{6}$$

The scaling factor is computed analytically as the ratio of the neighborhood's deformation trace to its original variance:

$$s_i = \frac{\text{Tr}(\Sigma_i)}{\sum_{j \in \mathcal{N}(i)} w_{ij} \|\mathbf{e}_{ij}\|^2} \tag{7}$$

In this phase, we followed a similar approach as the global alignment to determine the optimal hyperparameters on the same test set. Maintaining $\lambda_{\text{sim}} = 10^4$ from the previous phase, we evaluated $\lambda_{\text{ACAP}} \in 0.01, 0.1, 0.5, 1.0$. We selected $\lambda_{\text{ACAP}} = 0.5$, as it provided the best trade-off between reconstruction quality (CD) and alignment (NCC). One could straightaway train a learnable latent code $\mathcal{Z}$ and the MLP $f_{\theta_2}$ to predict the deformation field. However, in our experience, it got stuck in a local minima and performed worse in reconstructing the target surface.

## 4. Experiments

### 4.1. Data

This study leveraged longitudinal MRI of 92 preterm neonates scanned shortly after birth and again at term-equivalent age, acquired as part of the Developing Human Connectome Project (Edwards et al., 2022; Makropoulos et al., 2018). Imaging was performed using a 3T Philips Achieva scanner equipped with a 32-channel neonatal head coil. T2-weighted images were acquired using a fast spin-echo sequence with TR/TE = 12,000/156 ms, an

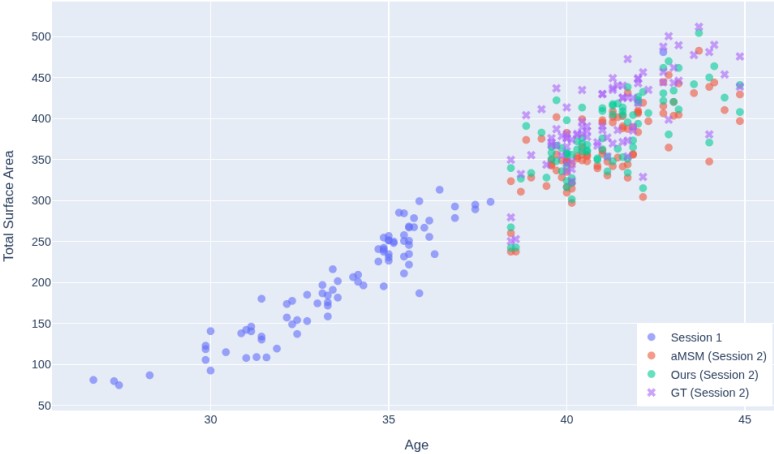

Figure 3: Total surface area of Session 1, Session-1 deformed surfaces(Ours), aMSM registered surfaces and unregistered ground truth surfaces. It shows that the total surface area is preserved more by the proposed method than the aMSM surfaces.

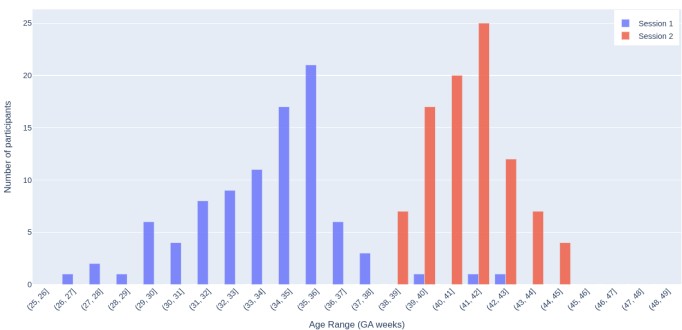

Figure 4: Distribution of scan ages for all 92 subjects: In blue the first scan taken shortly after birth, in orange the follow-up scan taken at around full term equivalent age.

in-plane resolution of 0.8 mm × 0.8 mm, a slice thickness of 1.6 mm, and a 0.8 mm overlap. The scans underwent motion correction and reconstruction (Makropoulos et al., 2018; Kuklisova-Murgasova et al., 2012; Cordero-Grande et al., 2018) before bias correction, brain extraction, and surface extraction using the fast, deep-learning-based cortical surface reconstruction pipeline (Ma et al., 2025). All surfaces were aligned to the MNI template using ANTS (Tustison et al., 2021) rigid registration and normalized for size to ensure consistent orientation and scale across all subjects. This normalization ensured that the computational model analyzed only meaningful structural differences. The surfaces were then resampled to a fifth-order icosphere (ico-5) of resolution 10242 vertices. Fig. 4 shows the scan age distribution across the dataset.

## 4.2. Baseline Methods

We compared our method to anatomical MSM (aMSM) (Robinson et al., 2017) to establish precise point-to-point correspondences between longitudinal cortical surface representations derived from the same subjects at distinct gestational time-points. The registration process was guided by the cortical curvature at the white matter surface. Using the standard aMSM protocol, mappings were initialized with a rigid, rotational alignment, followed by non-linear optimization performed in a coarse-to-fine hierarchical fashion. This optimization employed successively higher-resolution icospheric control point (CP) grids (e.g., *ico-2* through *order 4*), ensuring that initial alignment focused on prominent morphological features prior to the refinement of finer anatomical details, thereby encouraging smoothness. Furthermore, the mapping was regularised with a hyper-elastic strain-energy loss $E_{\text{STR}} = \sum_{\{p,q,r\}\in\mathcal{F}} W_{pqr}^2$, where:

$$W_{pqr} = \frac{\mu}{2}\left(R^k + R^{-k} - 2\right) + \frac{\kappa}{2}\left(J^k + J^{-k} - 2\right) \tag{8}$$

Here, $\{p,q,r\}$ represents each face vertex triplet; $J$ is the Areal Distortion it represents the relative change in area. $R$ is the Shape Distortion, which represents the relative change in the aspect ratio (shape) of the face; $\mu$ is the Shear Modulus, a physically-inspired parameter that penalizes changes in shape $(R)$; $\kappa$ is the Bulk Modulus, a physically-inspired parameter that penalizes changes in area $(J)$; and $k$ is a chosen integer (typically $k = 1$ or $k = 2$), which controls the stiffness of the penalty function. In practice we calculated $J$ and $R$ from the local affine deformation gradient 2D deformation gradient $\mathbf{F}_{pqr} \in \mathbb{R}^{2\times2}$ after projection into the tangent plane. Specifically

$$J = \lambda_1.\lambda_2 \quad R = \frac{\lambda_1}{\lambda_2} \tag{9}$$

Where, $\lambda_1$ and $\lambda_2$ represent the eigenvalues of $F_{pqr}$). The hyper-parameters $\mu$ and $\kappa$ were set to 0.4 and 1.6, to align with biomechanical models of cortical folding, and the tuning of regularisation verses similarity (calculated from cross correlation of curvature maps on the sphere) was empirically optimized across a representative subset of subjects spanning the gestational age distribution. The optimal $\lambda$ value was determined by analyzing the trade-off between feature correlation and the 95th percentile of estimated deformation strain, selecting a value situated at the limit of the linear relationship where further reduction in $\lambda$ yielded disproportionately high strain (distortion) relative to the marginal increase in feature similarity. For more details on the parameterisation of these experiments please reference (da Silva et al., 2025).

## 4.3. Training and optimisation

For training of the first phase, 64 subjects were used. 14 subjects for validation and 13 subjects for testing. Adam Optimizer with learning rate of $10^{-4}$ was used and trained for 1000 epochs. For instance optimization, all the pairs were used. Adam optimizer with learning rate of $10^{-4}$, reducing it at plateau by a factor of 0.5. We train for a total of 500 iterations per pair. Separate models were trained for the left and right hemispheres. For the first phase, it takes approximately 4 hours to train the network. Instance optimization takes 3 minutes per sample. aMSM usually takes 5-7 minutes per sample for registration. All experiments were performed on a single Nvidia 3090, 24GB GPU.

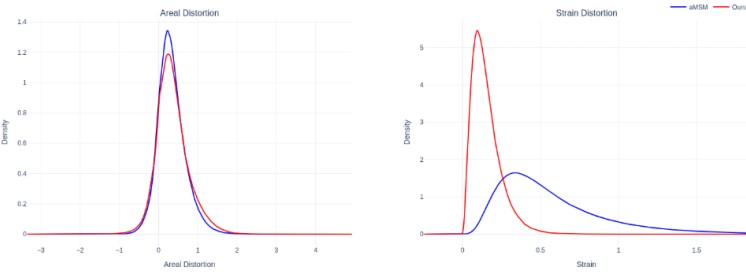

Figure 5: Areal and Shape Distortions for aMSM and the proposed method.

### 4.4. Evaluation Metrics

**Quantitative Metrics:** To quantitatively assess the performance of our proposed registration method, we employed several established metrics: we used the Bi-directional Chamfer Distance to quantify how well the deformed surfaces captured the anatomy of the target surface, and the Normalized Cross Correlation metric to quantify the alignment between the deformed and moving surfaces based on curvature.

Additionally, we evaluated preservation of surface topology by calculating the number of self-intersecting faces in the deformed surface, areal distortion and shape distortion(anisotropic strain). We also report areal and shape distortion (Eq 9) - calculated using connectome workbench commands (Marcus et al., 2011). Specifically to facilitate the reporting of results, we utilized $\log_2 |J|$ to quantify areal distortion and $\log_2 |R|$ to quantify shape distortion. Lower values correspond to reduced distortion levels in the deformed mesh. It is noteworthy that distortion is inherently non-zero for any deformed mesh, and a moderate degree of distortion is both acceptable and requisite for effective deformation. Accordingly, our evaluation of distortion included not only the mean and maximum values but also the 95th and 98th percentiles of $J$ and $R$, enabling comparison of distortion extents within physiological ranges across methods. To assess whether the deformed surfaces followed population level growth trends, we used the Desikan-Killiany-Tourville (DKT) parcellation extracted using the Melbourne Children's Regional Infant Brain Surface (MCRIB-S) pipeline (Alexander et al., 2019).

**Qualitative Evaluation:** We evaluate the smoothness of the growth maps, defined as the areal changes, which represent the expansion or contraction of cortical regions over time.

### 4.5. Results

Table 1 and Table 2 shows a comparison between aMSM and the proposed method. The proposed method improved the alignment over baseline while maintaining significantly lower shape distortion and similar areal distortion. Fig. 5 demonstrates the distortion density curves for the whole dataset. The proposed method had higher extreme areal distortion as indicated by the long tail, whereas shape distortion was significantly improved having narrower distribution. Fig. 1 shows smoother growth maps/areal expansion than the baseline with less number of small patches, which indicates a spatially smoother deformation of the cortical surface from one time point to another. In order to assess whether the areal distortion is leading to unwanted changes in the surface areas, we plotted the surface areas

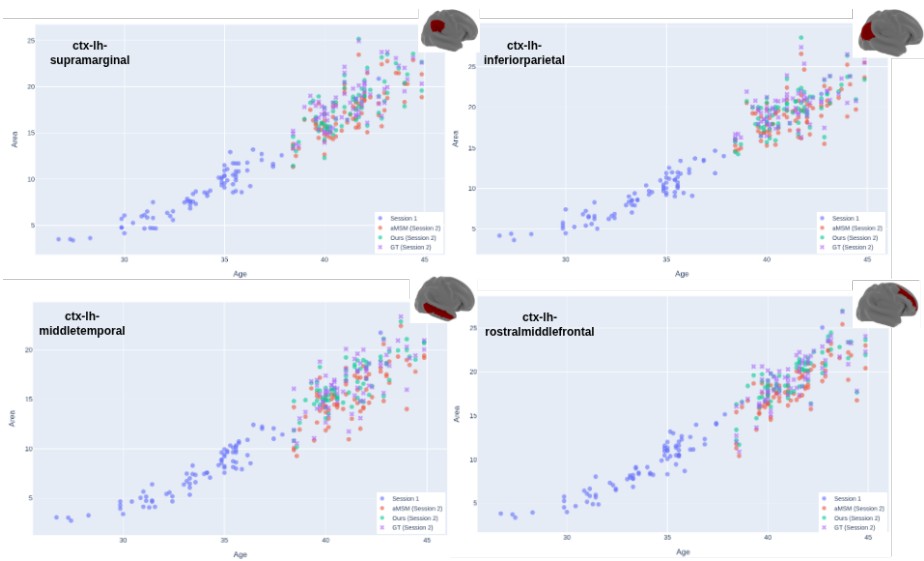

Figure 6: Scatterplots showing surface area of the parcellated regions of Session-1, Session-1 deformed surfaces (Ours), aMSM registered surfaces, and ground truth surface.

of multiple parcellated regions in Fig. 6. We plotted surface areas of Session1 surfaces which are used as moving surfaces, Session-1 deformed surfaces(Ours), aMSM registered surfaces and the ground truth unregistered surfaces. We saw that the surface areas were very close to the ground-truth surface areas suggesting no abnormally low or high deformations. Fig. 3 shows the total surface areas of Session-1 surfaces, Session-1 deformed surfaces(Ours), aMSM registered surfaces, and the ground truth surfaces. We saw that the total surface area was also maintained, with Ours being closer to the ground-truth surface areas.

Table 1: Comparison of Methods: Similarity and Areal Distortion metrics.

| Methods | Similarity | | Areal Distortion↓ | | | |
|---|---|---|---|---|---|---|
| | NCC↑ | CD↓ | Mean | Max | 95 % | 98 % |
| aMSM | $0.833 \pm 0.02$ | $4.124 \pm 0.2$ | $\mathbf{0.327 \pm 0.019}$ | $\mathbf{3.11 \pm 0.15}$ | $\mathbf{0.957 \pm 0.12}$ | $\mathbf{1.176 \pm 0.14}$ |
| Ours | $\mathbf{0.863 \pm 0.01}$ | $\mathbf{3.715 \pm 0.2}$ | $0.354 \pm 0.12$ | $4.92 \pm 0.22$ | $1.11 \pm 0.18$ | $1.365 \pm 0.11$ |

Table 2: Comparison of Methods: Shape Distortion and Self Intersection Faces (SIF).

| Methods | Shape Distortion↓ | | SIF %↓ |
|---|---|---|---|
| | Mean | Max | |
| aMSM | $0.571 \pm 0.26$ | $4.915 \pm 0.83$ | 0.0 |
| Ours | $\mathbf{0.153 \pm 0.003}$ | $\mathbf{2.55 \pm 0.33}$ | $0.004 \pm 0.00013$ |

## 5. Discussion

We propose a novel framework for longitudinal registration that operates directly on the 3D anatomical mesh, marking a significant departure from traditional methods that rely on spherical projection. Our experiments show that our method deforms the surface in a spatially smoother fashion than the baseline (Fig.1), resulting in a more clinically plausible growth of the cortex between any two time points. Clinical plausibility of the deformation field is further asserted by maintenance of the surface areas (Fig.6, Fig.3). By manipulating intricate anatomical surfaces in their native space, this approach offers a superior representation of true morphological changes. It establishes a foundation for comprehensive physics-based simulations capable of mechanistically predicting cortical deformation in contexts of development and pathology. Crucially, this methodology enables the forward modeling of cortical surfaces, facilitating the creation of interpretable, subject-specific growth models and the integration of biophysical constraints - capabilities that are unattainable through indirect spherical techniques. This, in turn, can enable us to understand how the cortical folds are formed throughout pregnancy to potentially discover pathological biomarkers leading to neurodevelopmental disorders. Building on this, future research will aim to explicitly model normative growth trajectories by adapting population-level growth maps to the individual. This can be done by simulating intermediate time points between any two scans of an individual, thereby advancing the development of subject-specific growth modeling within the native three-dimensional ambient space of the cortical surface. There are some limitations to the current method. In its current form, it only works on meshes with 10k vertices (ico5), which we intend to improve by extending it to a higher number of vertices and volumetric meshes to develop a comprehensive biomechanical forward model. This is possible since our framework's components (DiffusionNet and MLP) are discretization invariant and thus can be trained on a coarser mesh, but tuned or evaluated on a finer mesh.

## 6. Acknowledgement

We would like to acknowledge funding from the EPSRC Centre for Doctoral Training in Smart Medical Imaging (EP/S022104/1). Data were provided by the developing Human Connectome Project, KCL-Imperial-Oxford Consortium funded by the European Research Council under the European Union Seventh Framework Programme (FP/2007-2013) / ERC Grant Agreement no. [319456]. We are grateful to the families who generously supported this trial. CY/TSC/MFG were supported by NIH R01 MH060974.

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

# Appendix A. Qualitative Results

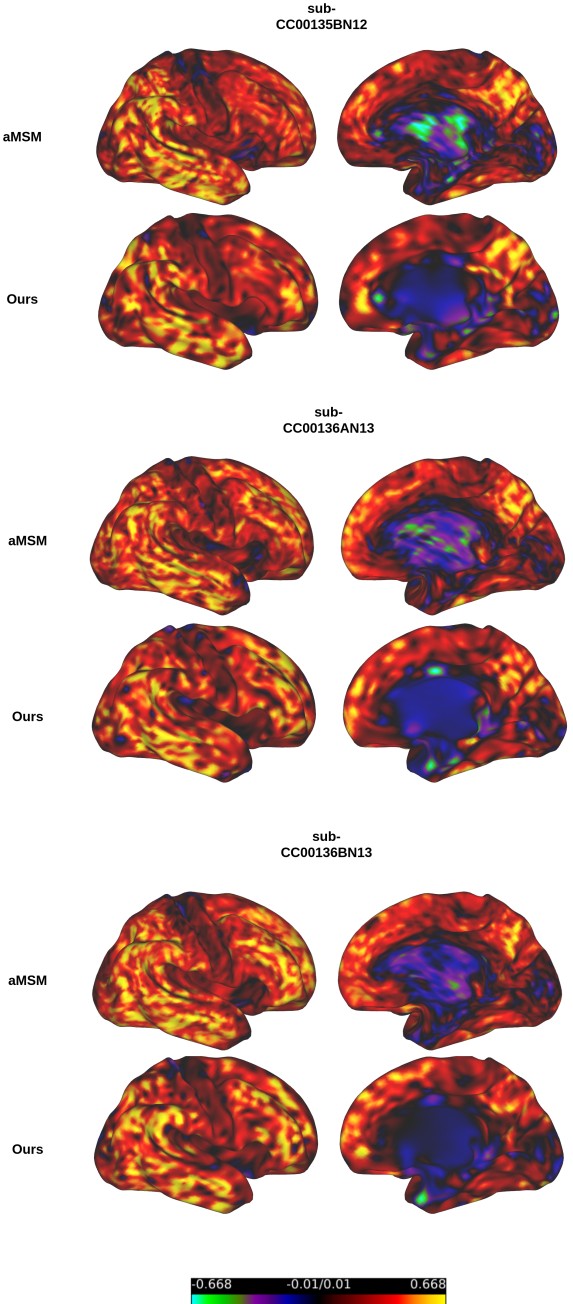

Figure 7: Growth Maps (Areal Distortions) shown for baseline (aMSM) and our method. Our method produces smoother and less patchy maps as compared to the baseline resulting in a spatially smoother growth of the cortex.

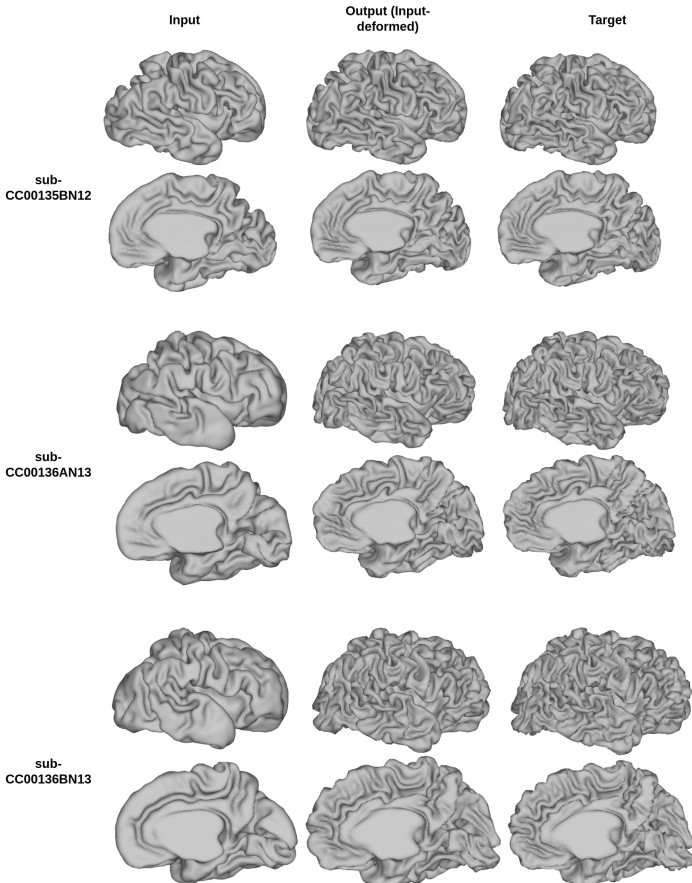

Figure 8: This figure shows the input surface to our model, the output deformed surface, and the target surface. As can be seen, the deformed surfaces are very close to the target surfaces.

We show more qualitative results from our experiments. Fig.7 shows that the growth maps from our methods are smoother. They are also less patchy than the baseline, suggesting a spatially smoother growth of the cortex from one time point to another. For the same subjects, we also show the input surface, the deformed surface from our method, and the target surface in Fig.8. The deformed surfaces are very close to the target surface, which is also suggested by a lower chamfer distance metric in table Tab.1. These results posit that our method can deform the input surface at one time point to a target surface at another time point while establishing vertex correspondence/alignment and maintaining a spatially smoother growth than the baseline.

## Appendix B. Sensitivity Analysis

Excessive regularization hindered learning large deformations, resulting in very smooth surfaces. The global optimization is highly sensitive to $\lambda_{NCC}$. Higher $\lambda_{NCC}$ led to smaller

deformations resulting in very smooth surfaces (high Chamfer Distance(CD))(Table 3). Note that NCC loss is only used in global optimization and thus the CD and NCC values in this table are different than the ones Table 1. $\lambda_{lap}$ and $\lambda_{NC}$ are more robust to their values, but without them, the number of self-intersection faces increases (Table 4, 5). $\lambda_{sim}$ proved to work the best with values of the order of $10^4$ (Table 7). The instance optimization is highly sensitive to $\lambda_{ACAP}$. Higher $\lambda_{ACAP}$ resulted in too much regularization, leading to smoother surfaces

The following table summarises the effect of different lambda values and the resulting reconstruction quality and alignment of the curvature metric.

Table 3: Performance metrics across different $\lambda_{NCC}$

| Methods | Similarity | |
|---|---|---|
| $\lambda_{\text{NCC}}$ | $\textbf{CD} \downarrow (\times 10^{-6})$ | $\textbf{NCC} \uparrow$ |
| 1 | 5.12 | 0.68 |
| 10 | 7.23 | 0.78 |
| 100 | 9.24 | 0.81 |

Table 4: Self Intersection Faces (SIF) across different $\lambda_{\text{lap}}$ values.

| Method | Metric |
|---|---|
| $\lambda_{\text{lap}}$ | $\textbf{SIF} \% \downarrow$ |
| 0 | 0.03 |
| 10 | 0.005 |

Table 5: Self Intersection Faces (SIF) across different $\lambda_{\text{NC}}$ values.

| Method | Metric |
|---|---|
| $\lambda_{\text{NC}}$ | $\textbf{SIF} \% \downarrow$ |
| 0 | 0.009 |
| 10 | 0.006 |

Table 6: Self Intersection Faces (SIF) for the combined $\lambda_{lap}$ and $\lambda_{NC}$ configuration.

| Method | Metric |
|---|---|
| $\lambda_{\text{lap}}/\lambda_{\text{NC}}$ | $\textbf{SIF} \% \downarrow$ |
| 10 / 10 | 0.004 |

Table 7: Cortical Distance (CD) across $\lambda_{sim}$.

| Method | Similarity |
|--------|------------|
| $\lambda_{\text{sim}}$ | $\mathbf{CD} \downarrow (\times 10^{-6})$ |
| $10^3$ | 4.1 |
| $10^4$ | 3.92 |
| $10^5$ | 3.715 |

