# OpenReview forum: "Anatomical Longitudinal Cortical Surface Registration"
_MIDL.io/2026/Conference — MIDL 2026 Poster_

### Official Review · Reviewer_zU48 · 2025-12-23

**Confidence:** 5
**Preliminary Rating:** 5
**Final Rating:** 5

**Summary:**

This paper proposes a framework for longitudinal (within-subject) cortical surface registration in neonate infants. The method comprises two stages: a DiffusionNet that learns vertex-wide shape features on a global scale (across all subjects), and a within-subject optimization step to refine higher frequency folding patterns. They perform experimental validation using a subset of the Developing Human Connectome project (n=92), and show that their method outperforms the baseline anatomical multimodal surface matching (aMSM) technique.

**Strengths:**

- The proposed method outperforms a similar, existing tool for deformable surface registration.
- The two stage protocol allows for global learning across the entire dataset and for within-subject refinement to obtain more accurate results.
- Their method circumvents the need for spherical projection by operating directly on the anatomical surface mesh (after rigid registration to MNI space).

**Weaknesses:**

There is little discussion of clinical significance regarding preterm, neonate data. Why did the authors select this specific application for their tool? Expanding on this slightly in the discussion and related works would be informative. The related works section states that "no framework has yet been able to capture the non-uniform dynamics of fetal and neonatal cortical folding", which identifies a gap in the literature, but it would be good to expand upon this in the introduction as well. The abstract makes no mention of fetal or neonate data.

**Detailed Comments:**

- The technical contributions of the paper address a need for improved neonate/fetal cortical surface registration and are well explained, but the paper should better highlight the clinical application.
- The short title was not properly defined (the header on all odd pages simply reads "short title").
- Page 8 has a reference that was not properly defined (the line just below equation 9).

**Justification Of Final Rating:**

The authors addressed all my concerns, which were minimal, and made improvements where necessary on an already well-done paper. For this reason, I am giving this manuscript this a 5 as my final rating.

**Justification Of The Preliminary Rating:**

The proposed method is novel and addresses a gap in the literature. The paper itself is well-written and nicely organized, although sometimes wordy (e.g., sentences are grammatically correct but difficult to follow due to their length).

**Questions To Address In The Rebuttal:**

- The paper makes no mention (that I can see) of cortical surface reconstruction algorithms specific to fetal or infant brains ex: Infant FreeSurfer). Would employing one of these methods improve results?
- I would like to see more discussion of the clinical implications in the introduction and discussion, especially because most readers will be less familiar with the challenges related to infant/fetal brain development than with aging/neurodegenerative data.

---

> ### Author Response · Authors · 2026-01-23
>
> We thank the reviewer for their valuable comments and insights.
>
>
> > **The paper makes no mention (that I can see) of cortical surface reconstruction algorithms specific to fetal or infant brains ex: Infant FreeSurfer). Would employing one of these methods improve results?**
>
> We respectfully disagree, Infant Freesurfer can only be utilized on T1-weighted brain MRIs from 0 to 2 year-olds.  Therefore, here we utilize Deep Learning based Neonatal surface reconstruction algorithm [1]  which is faster and have less metric distortion than freesurfer, and also works on T2-weighted perinatal brain MRIs.
>
> > **I would like to see more discussion of the clinical implications in the introduction and discussion, especially because most readers will be less familiar with the challenges related to infant/fetal brain development than with aging/neurodegenerative data.**
>
>
> During the perinatal development, the cortical surface experiences dramatic growth in the third trimeser,  characterised by formation of deep folds. This work creates a foundation for future subject-specific bio-mechanistic forward modelling of cortical surfaces. Enabling us to understand how the folds are formed to potentially discover biomarkers that may occur during the third trimester of preganancy leading to neurodevelopmental disorders. There are some limitations to current methods. In its current form, it only works on 10k vertices which we intend to improve by extending it to volumetric meshes to develop a comprehensive biomechanical forward model.

---

> > ### Comment · Reviewer_zU48 · 2026-01-23
> >
> > Thank you for responding my questions/concerns. Could you please point to where in the revised text you addressed these? I don't see any added discussion of clinical implications specific to neonatal data. I also do not see the updated ablation study, sensitivity analysis, or discussion of hyperparameters, etc. Is it possible the incorrect version was uploaded? It seems the only part of the text indicating an update is the second paragraph in the introduction.

---

> > > ### Author Response · Authors · 2026-01-31
> > >
> > > >**Thank you for responding my questions/concerns. Could you please point to where in the revised text you addressed these? I don't see any added discussion of clinical implications specific to neonatal data. I also do not see the updated ablation study, sensitivity analysis, or discussion of hyperparameters, etc. Is it possible the incorrect version was uploaded? It seems the only part of the text indicating an update is the second paragraph in the introduction.**
> > >
> > >
> > > We sincerely thank the Reviewer for their thoughtful response and appreciate the opportunity to clarify these points. We have revised the introduction and discussion sections and incorporated additional qualitative results in the Appendix during the extended deadline. Furthermore, we invite the Reviewer to consult our collective responses to Reviewer YyK9 and Reviewer WbcD, where we have detailed our hyperparameter tuning protocols, provided a comprehensive sensitivity analysis, and included an additional baseline comparison; we formally commit to integrating all these supplemental evaluations and methodological details into the final camera-ready version of the manuscript.

---

### Official Review · Reviewer_YyK9 · 2026-01-06

**Confidence:** 2
**Preliminary Rating:** 4
**Final Rating:** 4

**Summary:**

The paper proposes a longitudinal cortical surface registration framework that operates directly on anatomical meshes, combining a DiffusionNet-based encoder with a deformation MLP and per‑pair instance optimisation. The method is evaluated on preterm neonatal cortical surfaces from the dHCP. The approach aims to yield smoother growth maps, lower shape distortion, and better preservation of cortical surface area than traditional methods.

**Strengths:**

1. Addresses an important gap by performing longitudinal registration using deep learning methods instead of using traditional methods via spherical projections.
2. Results indicate improved similarity and substantially reduced shape distortion compared with aMSM

**Weaknesses:**

1. The paper is notation‑heavy, where several symbols and loss terms are introduced quickly and could benefit from a concise explanation included before discussing the proposed method.
2. Comparison is limited to a single baseline.
3. Some implementation and training details (choice of hyperparameters values, convergence behaviour of instance optimisation, training/inference times) are only briefly justified or tuned empirically without much sensitivity analysis

**Detailed Comments:**

1. When showing growth maps (Fig. 1), consider adding small annotations that directly link the observed differences to clinical interpretation.
2. Overall, the interpretability of the plots could be enhanced a lot by increasing the plot marker, text, and annotation size.

**Justification Of Final Rating:**

Thank you to the authors for their detailed responses and clarifications. While the manuscript has been meaningfully revised in line with all the reviewer comments,the mathematical exposition remains dense, limiting accessibility. Overall I would like to keep my original rating of weak acceptance, as further clarity is needed in the way the paper is presented.

**Justification Of The Preliminary Rating:**

Overall, this is a strong and timely paper that moves longitudinal cortical registration from spherical projections to direct anatomical meshes using a well‑designed hybrid learning/optimization framework. The main limitations lie in the relatively narrow set of baselines and the heavy notation, which could be streamlined and supplemented with clearer high‑level explanations of each component’s role.

**Questions To Address In The Rebuttal:**

1. How sensitive are your results to the chosen hyperparameters $\lambda_{sim}$, $\lambda_{NCC}$, $\lambda_{Lap}$, $\lambda_{NC}$, $\lambda_{ACAP}$? Did you observe regimes where excessive regularization degraded similarity, or vice versa, and could you summarize that sensitivity?
2. Could you comment on computational cost: approximate training time for the global model, average per‑pair instance‑optimization time, and how this compares practically to running aMSM?
3. For instance-optimisation, how often does optimization get stuck in poor local minima?
4. Could you explain the purpose of Figure 3? -  "Fig.3 shows the total surface areas of Session-1 surfaces, Session-1 deformed surfaces(Ours), aMSM registered surfaces, and the ground truth surfaces. We saw that the total surface area was also maintained, with Ours being closer to the ground-truth surface areas." It is difficult to understand what the plot is trying to show and what the expected pattern vs what we are observing.

---

> ### Author Response · Authors · 2026-01-23
>
> We thank the reviewer for their valuable comments. We are glad to reviewer finds our method addressing important gap in the domain that works.
>
> > **How sensitive are your results to the chosen hyperparameters λsim, λNCC, λLap, λNC, λACAP? Did you observe regimes where excessive regularization degraded similarity, or vice versa, and could you summarize that sensitivity?**
>
> Yes, excessive regularization hindered learning large deformations, resulting in very smooth surfaces. The global optimization is highly sensitive to lambda_NCC. Higher lambda_NCC led to smaller deformations resulting in very smooth surfaces (high Chamfer Distance(CD)). Lambda_lap and Lambda_NC are more robust to their values, but without them, number of self intersection faces  increased. Lambda_sim proved to work the best with values of the order of 10^5, and CD increased with lower values of lambda_sim
>
> The optimization is highly sensitive to lambda_ACAP. Higher lambda_ACAP resulted in too much regularization leading to smoother surfaces
>
> The following table summarises the effect of different lambda values and the resulting reconstruction quality and alignment of curvature metric.
>
> | $\lambda_{\text{NCC}}$ | CD ↓ ($\times 10^{-6}$) | NCC ↑ |
> | :--- | :--- | :--- |
> | 1 | 5.12 | 0.68 |
> | 10 | 7.23 | 0.78 |
> | 100 | 9.24 | 0.81 |
>
> Note that NCC loss is only used in global optimization and thus the CD and NCC values in this table  are different than the ones in the paper.
>
> | $\lambda_{\text{lap}}$ | SIF% ↓ |
> | :--- | :--- |
> | 0 | 0.03 |
> | 10 | 0.005 |
>
> ---
>
> | $\lambda_{\text{NC}}$ | SIF% ↓ |
> | :--- | :--- |
> | 0 | 0.009 |
> | 10 | 0.006 |
>
> ---
>
> | $\lambda_{\text{lap}}$ / $\lambda_{\text{NC}}$ | SIF% ↓ |
> | :--- | :--- |
> | 10 / 10 | 0.004 |
>
> ---
> | $\lambda_{\text{sim}}$ | CD ↓ ($\times 10^{-6}$) |
> | :--- | :--- |
> | $10^3$ | 4.1 |
> | $10^4$ | 3.92 |
> | $10^5$ | 3.715 |
>
> > **Could you comment on computational cost: approximate training time for the global model, average per‑pair instance‑optimization time, and how this compares practically to running aMSM?**
>
>
>
>
>
> Sorry for overlooking at this point, we will include this in appendix part of the paper:
>
>
>
> Training time:
>
> Global model: 4 hours
>
> Average per-pair instance optimization time: 3 minutes
>
> AMSM: 5 minutes
>
> Our model has to be trained with the whole dataset for 4 hours, and then it takes 3 minutes on an average for instance optimization.
>
>  > **For instance-optimisation, how often does optimization get stuck in poor local minima?**
>
>
>
> The model is generally robust to getting stuck in poor local minima. But we noticed that higher the age difference between the two scans, the higher the chances of achieving a sub-optimal solution. Usually running the optimization twice yields good results.
>
>
>
>
>
> > **Could you explain the purpose of Figure 3? - "Fig.3 shows the total surface areas of Session-1 surfaces, Session-1 deformed surfaces(Ours), aMSM registered surfaces, and the ground truth surfaces. We saw that the total surface area was also maintained, with Ours being closer to the ground-truth surface areas." It is difficult to understand what the plot is trying to show and what the expected pattern vs what we are observing.**
>
> We regard the first time point as session 1, and second time point as session 2 in our longitudinal data. Since session 1 precedes session 2, total surface area should exponentially increase from session 1 to session 2. We show the same in Fig3.  Session-1 deformed surfaces (Ours) is session 1 surface deformed into session 2 surface, and thus its surface area should increase and should be as close as possible to the GT session 2 surface area.
>
> aMSM registered surfaces, on the other hand, are the session 1 deformed surfaces by using the projected spherical warps. This deformed surface’s total surface area should also be close to the GT session 2 surface.
>
> We see that our method preserves the session 2 surface area better than the aMSM.

---

### Official Review · Reviewer_WbcD · 2026-01-10

**Confidence:** 4
**Preliminary Rating:** 2

**Summary:**

The paper proposes a method for registering cortical surfaces. More specifically, it proposes a deep learning framework enhanced with iterative optimization that operates on the brain's anatomical mesh, avoiding the need for projection to spherical coordinates. The method yields a smoother map compared to the baseline, while preserving the topological integrity of the original data.

**Strengths:**

The paper discusses an interesting idea of avoiding the projection of the cortical surface to spherical coordinates. This could potentially alleviate the method from further introduction of noise and errors while being, by construction, more interpretable.
It also proposes a nice architecture that combines feature extraction and training to recover roughly the global registration, with registration refinement achieved through iterative optimization, thereby combining the best of both worlds.
Finally, it attempts to address the challenging issue of cortical surface registration on neonatal data, which lacks many longitudinal points.

**Weaknesses:**

1. Although I believe that the method is interesting and it has merits, I found that the text needs some brushing up. For example there are typos e.g. Abstract: "In this work" or syntax errors 1st paragraph of page 2 "However, spherical ...".
2. From the intro, contributions, and related work, it is not very clear how the proposed work is different from some of the state-of-the-art methods. I believe the authors need to revisit their argumentation and focus on what makes their work distinct from previous methods, especially those that do not project to spherical coordinates and those that use DL with an iterative optimization combination.
3. I like the combination of feature extraction and the iterative optimization scheme. I would like to further as why the authors chose this specific extractor and whether they have tried other ones. Is there any dissadvantage in choosing this one?
4. Similarly, why did the authors choose an MLP instead of just optimizing the transformation parameters, or using a transformer or any other network? I believe that justifying these choices would improve the paper quality.
5. I am not sure if I missed it, but do the authors describe anywhere how to tune their hyperparameters and the ones from the baseline?
6. I found the experimentation rather limited. The paper just compares with one baseline. I believe it would be nice to see at least one method that also performs DL with iterative optimization and potentially a geometric DL one(?).
7. I would like to ask the authors to comment more on why they believe that the areal distortion of the baseline is better than the proposed method and whether they believe that this might be a limiting factor when someone wants to use their method. In this train of thought, I believe that maybe some discussion about limitations of future work would also increase the paper quality.
8. Is the baseline also trained with NCC? If not I am not sure whether the comparison of the proposed method trained with NCC is fair when it comes to NCC itself.
9. In Table 1, could the authors explain what SIF means?
10. The code is not provided until acceptance.
11. In Appendix B, the proof is missing.
12. It would be nice to see some qualitative results as well.

**Detailed Comments:**

Please see the weaknesses.

**Justification Of The Preliminary Rating:**

The paper presents is a conceptually interesting approach and introduces a new combination of DL with iterative optimization for cortical surface registration to operate directly on the surface points avoiding projection to the spherical coordinates.

However, I believe that the limitations in the evaluation pipeline and the way the method is advertised make the paper’s quality insufficient for acceptance in MIDL this year. I believe the authors have to put substantial effort into enhancing the motivation, the literature review, and supporting their storyline with more experiments, as well as including more baselines and qualitative results. Additionally, the discussion needs to be revised to incorporate limitations and future work.

Given these shortcomings, I recommend rejection in its current form. Nonetheless, the underlying idea is promising, and I encourage the authors to strengthen the paper and resubmit it to a future venue.

**Questions To Address In The Rebuttal:**

I would like to invite the authors to revisit their work as a whole. The paper would benefit from some polishing. Moreover the incorporation of baselines is critical and the sharpening of the argumentation of what the paper proposes.

---

> ### Author Response · Authors · 2026-01-23
>
> We thank the reviewer for their valuable comments, and we are glad that reviewers found our work interesting and share our sentiment that the proposed method more robust and interpretable. Now we address all the queries individually:
>
>
>
> > **From the intro, contributions, and related work, it is not very clear how the proposed work is different from some of the state-of-the-art methods. I believe the authors need to revisit their argumentation and focus on what makes their work distinct from previous methods, especially those that do not project to spherical coordinates and those that use DL with an iterative optimization combination.**
>
>
>
> Thanks for pointing this out. We have now edited the introduction to better highlight our novelty in our contribution. Mainly, the work is different because it seeks to solve a fundamentally different problem from most other cortical alignment frameworks: simulation of perinatal (fetal and neonatal) cortical growth and folding, which requires large deformations.; whereas other frameworks are either primarily designed to optimise spatial mappings between different brains (MSM[8], spherical demons[9], DDR[7]) by projecting them onto a sphere or by combining surface reconstruction followed by longitudinal correspondences on native anatomy between aging brains, which is a much more subtle process [2]. Our work is distinct from current SOTA methods, as we operate only in 3D ambient space rather than spherical projection, focusing on dramatic changes in perinatal age, and laying a foundation for prescribing the biomechanics of deformations for forward modelling.
>
> > **I like the combination of feature extraction and the iterative optimization scheme. I would like to further as why the authors chose this specific extractor and whether they have tried other ones. Is there any disadvantage in choosing this one?**
>
>
>
> We are glad that the reviewer likes our approach of combining both feature extraction and iterative optimization. More specifically, Cortical surfaces are very convoluted and usually represented with high number of vertices (~130k). The computational needs of most geometric deep learning frameworks (e.g. Pointnets and surface transformers) increase significantly with the number of input nodes . For 10K vertices it takes 30GB of VRAM to encode meshes with batch size 1 whereas DiffusionNet takes 2GB of VRAM.  Moreover, we found it to be twice as fast to train as compared to surface transformer[11]. While the current framework was validated at 10k, future progression towards forward modelling will involve training on volumetric finite element meshes at much higher resolution  e.g this recent FEM paper builds a model from the same data based on a 160k vertex mesh [9]. DiffusionNet helps with transitioning to such a high number of vertices, as it is a fundamentally discretization-invariant method. Furthermore, evidence of success in various mesh-based learning tasks, including surface parcellation[3], neural deformation field [4], shape matching [5] made us choose this architecture.
>
>
> > **Similarly, why did the authors choose an MLP instead of just optimizing the transformation parameters, or using a transformer or any other network? I believe that justifying these choices would improve the paper quality.**
>
>
>
> Thanks for pointing the missing justification, we will include this in the paper. We have provided the justification MLP later, however transformer architecture is also a valid choice which requires more careful optimization strategies, which we leave to explore in the future work :
>
> We aim to create a discretization invariant method that can be trained on coarser meshes (less number of vertices) and can be generalized to finer meshes (high number of vertices). MLP as a deformation network serves the purpose as it is invariant to the input grid structure and can represent a continous 3D field. Furthermore, we use MLP as its a simple network which is easy to debug and build upon.

---

> ### Author Response · Authors · 2026-01-23
>
> > **I am not sure if I missed it, but do the authors describe anywhere how to tune their hyperparameters and the ones from the baseline?**
>
> Thanks for pointing this out, we will discuss this in the paper. To answer your question, to optimize the hyperparameters of the proposed method, we performed an empirical evaluation using a subset of the data evenly spanning the gestational age, consisting of 14 samples for validation and 13 for testing. We evaluated a range of values for the loss function weights across two phases.
>
> In the first phase, we trained multiple models with:
> $\lambda_{\text{sim}} \in \{10^3, 10^4, 10^5\}$, $\lambda_{\text{NCC}} \in \{1, 10, 100\}$, $\lambda_{\text{lap}} \in \{1, 10, 100\}$, $\lambda_{\text{NC}} \in \{1, 10, 100\}$
>
> Based on achieving the lowest reconstruction error, measured by Chamfer Distance (CD), and the highest alignment, measured by Normalized Cross-Correlation (NCC), we selected $\lambda_{\text{sim}} = 10^4$, and $\lambda_{\text{NCC}} = \lambda_{\text{lap}} = \lambda_{\text{NC}} = 10$.
>
> In the second phase, we followed a similar approach to determine the optimal hyperparameters via instance-optimization on the same test set. Maintaining $\lambda_{\text{sim}} = 10^4$ from the previous phase, we evaluated $\lambda_{\text{ACAP}} \in \{0.01, 0.1, 0.5, 1.0\}$. We selected $\lambda_{\text{ACAP}} = 0.5$, as it provided the best trade-off between reconstruction quality (CD) and alignment (NCC).
>
> > **I found the experimentation rather limited. The paper just compares with one baseline. I believe it would be nice to see at least one method that also performs DL with iterative optimization and potentially a geometric DL one(?).**
>
> We see the reviewer's point on this, but to the best of our knowledge, there aren’t any research utilizing the anatomical meshes for cortical surface registration for neonatal data, either with DL+instance optimization or based on geometric DL. We would be very happy to include experiments in camera-ready, if reviewers could point to other baseline methods.
>
> However, we choose to use a widely adopted and validated aMSM for comparison, which is a per-instance iterative method and is closest to our methodology. There are learning based generalizable DL based spherical projection methods like DDR[7], Sugar[6]. We trained and evaluated DDR due to its competitive performance and easily available code, using the same train,valid and test split of the neonatal dataset and report the following results. We also report the mean and std of all three methods across 3 runs each.
>
> | Methods | NCC ↑ | CD ↓ | Areal Dist: Mean ↓ | Areal Dist: Max ↓ | Areal Dist: 95% ↓ | Areal Dist: 98% ↓ | Shape Dist: Mean ↓ | Shape Dist: Max ↓ | SIF % ↓ |
> | :--- | :--- | :--- | :--- | :--- | :--- | :--- | :--- | :--- | :--- |
> | **aMSM** | 0.833 ± 0.02 | 4.124 ± 0.2 | **0.327 ± 0.019** | 3.11 ± 0.15 | 0.957 ± 0.12 | 1.176 ± 0.14 | 0.571 ± 0.26 | 4.915 ± 0.83 | 0.0 ± 0 |
> | **Ours** | **0.863 ± 0.01** | **3.715 ± 0.2** | 0.354 ± 0.012 | 4.92 ± 0.22 | 1.11 ± 0.18 | 1.365 ± 0.11 | **0.153 ± 0.003** | 2.55 ± 0.33 | 0.004 ± 0.00013 |
> | **DDR** | 0.84 ± 0.02 | 4.22 ± 0.1 | 0.36 ± 0.03 | **0.78 ± 0.31** | **0.62 ± 0.42** | **0.88 ± 0.23** | 0.71 ± 0.11 | **1.5 ± 0.67** | **0.002 ± 0.0001** |
>
> > **I would like to ask the authors to comment more on why they believe that the areal distortion of the baseline is better than the proposed method and whether they believe that this might be a limiting factor when someone wants to use their method. In this train of thought, I believe that maybe some discussion about limitations of future work would also increase the paper quality**
>
> One of the reasons for the areal distortion of the baseline to be better is that the baseline (aMSM) works on a discrete optimization framework where it only moves the vertices on the sphere to a limited set of very close candidate points. Whereas the proposed method has the freedom to move the vertices to anywhere, which is modulated by the regularizers. Since there is no ground truth on the areal distortion between any two surfaces, we hypothesise that as long as the areal distortion are within the biological limits, and the growth maps are smooth (FIg,1 in the paper), the method has the utility for further research.

---

> > ### Comment · Reviewer_WbcD · 2026-01-27
> >
> > >We see the reviewer's point on this, but to the best of our knowledge, there aren’t any research utilizing the anatomical meshes for cortical surface registration for neonatal data, either with DL+instance optimization or based on geometric DL. We would be very happy to include experiments in camera-ready, if reviewers could point to other baseline methods.
> >
> > I would like to further ask the authors whether they can use a method that utilizes anatomical meshes, but not necessarily for neonatal data. What are the caveats?
> >
> > Furthermore, I would like to thank the authors for their efforts and their responses. However, I believe more work is needed to improve the story and discussion, and the qualitative results are still missing. I would like to remain in my original score and encourage the authors to improve the quality of the paper and submit it to a future venue.

---

> > > ### Author Response · Authors · 2026-01-31
> > >
> > > We thank the reviewer for his response and answer his question.
> > > >**I would like to further ask the authors whether they can use a method that utilizes anatomical meshes, but not necessarily for neonatal data. What are the caveats?**
> > >
> > > While we acknowledge the contributions of V2C-long [2] in utilizing anatomical meshes for longitudinal registration(by formulating it as a surface reconstruction problem), its application is primarily optimized for aging brain populations, such as those in the ADNI dataset, where cortical surface changes are characterized by subtle, slow-acting **atrophy(shrinkage)**. In contrast, our work addresses the unique challenges of the neonatal developing brain, which undergoes rapid, dramatic, and non-uniformly distributed **expansion** that necessitates a framework capable of modeling large-scale deformations. By explicitly formulating this as a surface deformation problem rather than a reconstruction task, we provide a systematic foundation for integrating soft-tissue mechanics, which we believe is essential for capturing the complex biomechanical dynamics of cortical development.
> > > Utilizing models that were trained on aging-related shrinkage may not adequately predict large expansion of cortical development.
> > >
> > > >**Furthermore, I would like to thank the authors for their efforts and their responses. However, I believe more work is needed to improve the story and discussion, and the qualitative results are still missing. I would like to remain in my original score and encourage the authors to improve the quality of the paper and submit it to a future venue.**
> > >
> > > We sincerely thank the reviewer for their constructive feedback. Sharing the reviewer’s commitment to improving the work’s rigor, we have revised the introduction and discussion sections in the manuscript, as well as added new qualitative results in the Appendix during the extended deadline. We have reported several key enhancements: specifically, we added an additional baseline comparison, reported the mean and standard deviation across three independent runs for all methods, and provided comprehensive details on hyperparameter optimization (previous comments). Furthermore,  we would like to direct the reviewer to our response to Reviewer YyK9, which contains a detailed sensitivity analysis and an elaboration on training and testing run times. We believe these updates, prompted by the reviewer's valuable suggestions, have significantly strengthened the manuscript. We completely agree with the reviewer's views and will add these changes to the camera-ready version of the manuscript.

---

> ### Author Response · Authors · 2026-01-23
>
> > **Is the baseline also trained with NCC? If not I am not sure whether the comparison of the proposed method trained with NCC is fair when it comes to NCC itself.**
>
> Reviewer is indeed correct, the baseline is trained with NCC. We will make sure to highlight this
>
>
> > **In Table 1, could the authors explain what SIF means?**
>
> Sorry for this, SIF: Self Intersection faces. Its the number of intersecting faces.
>
>
> > **It would be nice to see some qualitative results as well.**
>
> We agree with the reviewers perspective, we are working on more visual results and will include in the camera ready.
>
>
>
>  -----
>
>
> [1] Lin Tian, Hastings Greer, Ra´ul San Jos´e Est´epar, Roni Sengupta, and Marc Niethammer.NEPHI: Neural deformation fields for approximately diffeomorphic medical image registration.
>
> [2] Fabian Bongratz, Jan Fecht, Anne-Marie Rickmann, and Christian Wachinger. V2c-long: Longitudinal cortex reconstruction with spatiotemporal correspondence. Imaging Neuroscience, 3, 01 2025. ISSN 2837-6056.
>
> [3] Zhu, Yuanzhuo, Chunfeng Lian, Xianjun Li, Fan Wang, and Jianhua Ma. 2024. ‘Efficient Cortical Surface Parcellation via Full-Band Diffusion Learning at Individual Space’. In Medical Image Computing and Computer Assisted Intervention – MICCAI 2024, edited by Marius George Linguraru, Qi Dou, Aasa Feragen, et al. Springer Nature Switzerland. https://doi.org/10.1007/978-3-031-72069-7_16.
>
> [4] Cha, Sihun, Serin Yoon, Kwanggyoon Seo, and Junyong Noh. 2025. ‘Neural Face Skinning for Mesh-Agnostic Facial Expression Cloning’. arXiv:2505.22416. Preprint, arXiv, May 28. https://doi.org/10.48550/arXiv.2505.22416.
>
> [5] Zhuravlev, Aleksei, Zorah Lähner, and Vladislav Golyanik. 2025. ‘Denoising Functional Maps: Diffusion Models for Shape Correspondence’. arXiv:2503.01845. Preprint, arXiv, April 2. https://doi.org/10.48550/arXiv.2503.01845.
>
> [6] Ren, Jianxun, Ning An, Youjia Zhang, et al. 2024. ‘SUGAR: Spherical Ultrafast Graph Attention Framework for Cortical Surface Registration’. Medical Image Analysis 94 (May): 103122. https://doi.org/10.1016/j.media.2024.103122.
>
> [7] Suliman, Mohamed A., Logan Z. J. Williams, Abdulah Fawaz, and Emma C. Robinson. 2022. ‘A Deep-Discrete Learning Framework for Spherical Surface Registration’. In Medical Image Computing and Computer Assisted Intervention – MICCAI 2022, edited by Linwei Wang, Qi Dou, P. Thomas Fletcher, Stefanie Speidel, and Shuo Li. Springer Nature Switzerland. https://doi.org/10.1007/978-3-031-16446-0_12.
>
> [8] Yeo, B. T. Thomas, Mert R. Sabuncu, Tom Vercauteren, Nicholas Ayache, Bruce Fischl, and Polina Golland. 2010. ‘Spherical Demons: Fast Diffeomorphic Landmark-Free Surface Registration’. IEEE Transactions on Medical Imaging 29 (3): 650–68. https://doi.org/10.1109/TMI.2009.2030797.
>
> [9] Robinson, Emma C., Kara Garcia, Matthew F. Glasser, et al. 2018. ‘Multimodal Surface Matching with Higher-Order Smoothness Constraints☆’. NeuroImage 167 (February): 453–65. https://doi.org/10.1016/j.neuroimage.2017.10.037.
>
> [10] Osman, Besm, Ruben Vink, Andrei Jalba, Kurt G. Schilling, and Maxime Chamberland. 2025. ‘A Subject-Specific Reversible Folding Model Reveals Geometry-Driven White-Matter Organization’. bioRxiv: The Preprint Server for Biology, December 12, 2025.12.10.693407. https://doi.org/10.64898/2025.12.10.693407.
>
> [11] Dahan, Simon, Logan Z. J. Williams, Daniel Rueckert, and Emma C. Robinson. 2024. ‘The Multiscale Surface Vision Transformer’. arXiv:2303.11909. Preprint, arXiv, June 11. https://doi.org/10.48550/arXiv.2303.11909.

---

### Author Rebuttal · Authors · 2026-01-23

**Rebuttal:**

In this revised manuscript, we have updated the Introduction and Discussion sections, emphasising the clinical utility, future works, and limitations. We have also added more qualitative results in the Appendix (Reviewer-1 - WbcD). Furthermore, in response to the review process, we have incorporated new hyper-parameter sensitivity analysis (Reviewer-2 -YyK9), expanded the discussion on hyperparameters and modeling choices(Reviewer-1 - WbcD), clarified the clinical implications of our method (Reviewer-3-zU48, and Discussion Section of the manuscript), added one more method comparison(Reviewer-1 - WbcD), and responded to all reviewers' concerns. We will add all the new analyses (mentioned as comments) in the camera-ready version of our manuscript.

**Supporting Material:**

/attachment/4848b419cbd17e7476cdee4f2161da34c4698627.pdf

---

### Meta-Review · Area_Chair_JwK1 · 2026-02-06

**Recommendation:** Accept (Poster)
**Confidence:** 4

**Metareview:**

This is a borderline paper with reviewers having different opinions.

The paper proposes a hybrid Deep Learning and instance optimization method for neonatal/preterm cortical surface registration.
The main novelty of the method is the combination of a Deep Learning approach for rough global registration and a subsequent instance optimization method that includes biomechanical regularization.
The reviewers agree that this approach is novel and interesting.

The weaknesses of this paper are the experimental validation, where they initially compared to only one baseline method aMSM, in the discussion period they added a Deep Learning baseline DDR - which is still missing from the manuscript.
The authors argue that other learning methods are mostly focused on cortical surface reconstruction with application in neurodegenerative diseases and aging. It still would be interesting to add a comparison to those methods - but I understand this was out of the scope for this rebuttal.

I lean towards accepting this paper as a poster presentation, with the main reason being the interesting combination of learning based registration and biomechanically constrained instance optimization. I believe the method is interesting to the community and can be further validated and extended in future work. The authors should add the additional results presented in the discussion period to the camera ready paper/ appendix.

---

### Decision · Program_Chairs · 2026-02-14

Accept (Poster)